

# Trends in the atmospheric water vapour estimated from GPS data for different elevation cutoff angles

Tong Ning[1] and Gunnar Elgered[2]

[1]Lantmäteriet (The Swedish Mapping, Cadastral and Land Registration Authority), SE-80182, Gävle, Sweden
[2]Department of Space, Earth and Environment, Chalmers University of Technology, Onsala Space Observatory, SE-43992 Onsala, Sweden.

*Correspondence to:* T. Ning (tong.ning@lm.se)

**Abstract.** We have processed 20 years of GPS data from 8 sites in Sweden and 5 sites in Finland, using two different elevation cutoff angles 10° and 25°, to estimate the atmospheric integrated water vapour (IWV). We have also tested three additional elevation-angle-dependent parameters in the GPS data processing, i.e., (1) two different mapping functions, (2) with or without second order corrections for ionospheric effects, and (3) with or without elevation dependent data weighting. The results show that all these three parameters have insignificant impacts on the resulting linear IWV trends. We compared the GPS-derived IWV trends to the corresponding trends from radiosonde data at 7 nearby ($< 120$ km) sites and the trends inferred from the European Centre for Medium-Range Weather Forecasts (ECMWF) reanalysis data (ERA-Interim). The IWV trends given by GPS elevation 10° and 25° solutions show similar results when compared to the trends from the nearby radiosonde data, with correlation coefficients of 0.71 and 0.74, respectively. In addition the 25° solution gives a slightly lower root-mean-square (RMS) difference (0.15 kg/(m$^2$·decade)) than the 10° solution (0.17 kg/(m$^2$·decade)). When compared to the IWV trends obtained from ERA-Interim, the GPS solution for the 25° elevation cutoff angle gives a higher correlation (0.90) and a lower RMS difference (0.09 kg/(m$^2$·decade)) than the ones obtained for the 10° solution (0.53 and 0.18 kg/(m$^2$·decade)). The results indicate that a higher elevation cutoff angle is meaningful when estimating long term trends, and that the use of different elevation cutoff angles in the GPS data processing is a valuable diagnostic tool for detection of any time varying systematic effects, such as multipath impacts.



# 1  Introduction

Atmospheric water vapour is a very important greenhouse gas due to its ability of absorbing long wave thermal radiation emitted from the Earth's surface. A warmer climate increases the amount of water vapour, which further reduce the amount of long-wave radiation escaping from the Earth to space, and thereby leading to an even warmer climate. Hence, the atmospheric water vapour is one of

the most significant climate feedback processes and therefore atmospheric integrated water vapour (IWV) can be used as an independent data source for monitoring climate changes. This, however, requires accurate observations with a long-term stability in order to have a high accuracy of the estimated trends in the IWV. With a relatively high temporal resolution, continuously improving spatial density, and less expensive receivers, ground-based GNSS networks have been identified as a

useful technique to obtain long-term trends in the atmospheric IWV [e.g., *Wang et al.* 2007, *Nilsson and Elgered* 2008, and *Vey et al.* 2010].

A reliable estimate of the IWV trend requires homogeneous input data. The observations acquired from the ground-based GNSS stations may, however, contain inconsistencies which need to be thoroughly investigated and corrected, if it is possible, before GNSS-derived IWV are used for climate

monitoring. *Ning et al.* (2016a) pointed out that these are mainly due to systematic errors and cannot be averaged out as the time series of the measurements becomes longer. The origin of these errors can either be related to data processing or to hardware changes or the electromagnetic environment at the antenna site (*Vey et al.*, 2009). The first type of errors can be significantly reduced after a homogenous data reprocessing over the whole time series (*Steigenberger et al.*, 2007). Hard-

ware changes are normally well documented. For example, for most of the continuously operating GNSS stations, e.g., in the International GNSS Service (IGS) network (http://www.igs.org) and the Tide Gauge Benchmark Monitoring (TIGA) project (*Schöne et al.*, 2009), there are archived log files with a common format to record all hardware changes. This is normally done by the owners of the GNSS sites, where a continuously updated log file for each station is provided (*Ning et al.*, 2016b).

Such log files are very important for the future homogenization of the data.

Changes in the electromagnetic environment result in different multipath effects on the GNSS observations. In difference to the hardware changes, which typically result in an instant constant offset, the inconsistences caused by multipath effects are normally not fixed in time but varies with the reflective properties, e.g., growing vegetation (*Pierdicca et al.*, 2014) and/or different soil mois-

ture (*Larson et al.*, 2010). Such changes are difficult to quantify and document. After examination of 19 years of data obtained at 101 TIGA stations, *Ning et al.* (2016b) found that about 70 % of the detected inconsistences in the GNSS-derived IWV time series cannot be related to any documented hardware change. Some of these detections may be false but the inconsistences in the IWV time series induced by multipath effects need to be eliminated, or at least significantly reduced.

In GNSS data processing, low elevation angles are used in order to improve the geometry which reduces the formal error of the individual IWV estimate. This is an advantage when we use the in-





ferred IWV for validation of specific observations over short time scales. For example, the ground-based GNSS IWV was identified as a priority 1 measurement for GCOS (Global Climate Observing System) Reference Upper Air Network (GRUAN) (*Ning et al.*, 2016a). However, if we use the
IWV estimates for monitoring the long-term changes in the atmospheric water vapour, e.g., as linear trends, the formal error of the individual IWV is not the limiting factor (*Nilsson and Elgered*, 2008). In this case, the homogeneity of the IWV time series is more important where higher elevation cutoff angles may be desired if signal multipath effects cannot be modelled accurately for observations at low elevation angles. Using 14 years of GPS data from 12 sites in Sweden and
Finland, *Ning and Elgered* (2012) found that a higher elevation cutoff angle (25°) gave the best agreement between the GPS-derived IWV trends and the ones obtained from radiosonde profiles at nearby launching sites.

There are also parameters used in GNSS data processing which are elevation-angle-dependent and therefore important to investigate. We know that they introduce systematic errors in terms of
biases. When studying trends the question is if these biases vary over very long time scales and thereby affecting the accuracy of estimated trends. Mapping functions (MFs) are used to convert the slant path delay to the equivalent zenith total delay (ZTD). The MF can induce significant errors in the slant delays when the elevation angle is low (*Stoew et al.*, 2007). The ionospheric delay is dependent on the total amount of free electrons along the propagation path, named total electron
content (TEC), and a larger ionospheric impact is expected for the GNSS signals coming from lower elevation angles. The standard correction is to form an ionosphere-free linear combination which can eliminate 99.9 % of the total ionospheric delay. However the contributions from higher-order terms can still have a significant impact during strong solar activities (*Pireaux et al.*, 2010). In addition an elevation-dependent data weighting is sometimes also used. It is actually recommended for the data
processing of the EUREF permanent network (*Huber and Kaniuth*, 2003). When the elevation cutoff angle is lower than 20°, *Huber and Kaniuth* (2003) found better repeatabilities of the daily north and east position components, over an analysed time period of ten days, after applying elevation-dependent weighting functions in the GPS data processing.

We carried out a study with a focus on the elevation-angle-dependent parameters and their corre-
sponding impacts on the resulting IWV linear trends. The IWV estimated from the GPS data were compared to the ones obtained from radiosondes and ERA-Interim data. The data sets are described in Section 2 where details about the GPS data processing are given. The primary interest of this work is if there are multipath, or other antenna environment, effects on the estimated IWV. Therefore any known interventions in the GPS observations due to hardware changes need to be assessed.
This is discussed in Section 3 which also gives details on the estimation of IWV trends. Section 4 presents the comparisons of the GNSS-derived IWV with the ones obtained from the different data sets. Finally, the conclusions are given in Section 5.



## 2 Data sets

### 2.1 GPS

We have analysed 20 years (from 1 January 1997 to 31 December 2016) of ground-based GPS observations acquired from 8 sites in the Swedish reference network (SWEPOS) and 5 sites in the Finnish reference network (FinnRef). Figure 1 depicts the location of the sites and their coordinates are given in Table 1.

A standard data processing was first carried out using GIPSY/OASIS II v.6.2 (*Webb and Zumberge*, 1993) with the Precise Point Positioning (PPP) strategy (*Zumberge et al.*, 1997). The inputs of the processing were ionospheric free linear combinations formed by acquired GPS phase-delay observations while the output included station coordinates, clock biases, and tropospheric parameters. The final GPS orbit and clock products used were provided by Jet Propulsion Laboratory (http://gipsy.oasis.jpl.nasa.gov/gipsy/docs/GipsyUsersAGU2007.pdf). An ocean tide loading correction using the FES2004 model (*Lyard et al.*, 2006) was applied while no atmospheric pressure loading corrections were used. The absolute calibration of the Phase Centre Variations (PCV) for all antennas (from the file igs08_1869.atx) was implemented (*Schmid et al.*, 2007) and the technique of ambiguity resolution was applied (*Bertiger et al.*, 2010).

The ZTD and linear horizontal gradients were estimated using a random walk model with a standard deviation (SD) of 10 mm/$\sqrt{\text{h}}$ and 0.3 mm/$\sqrt{\text{h}}$, respectively. The SD value used for the ZTD is given by *Jarlemark et al.* (1998) where they found a temporal variability in the wet delay, derived from 71 days of microwave radiometer measurements, varying in the interval 3–22 mm/$\sqrt{\text{h}}$ at the Onsala site. This value however is larger than the one given by http://acc.igs.org/workshop2016/presentations/Plenary_05_03.pdf where they recommend a SD value of 3 mm/$\sqrt{\text{h}}$ when applying an equal weighting. In order to investigate possible impacts of using a larger SD value on the resulting IWV trends we have processed data again for two sites (SODA and OVE0) which are located far north (expecting a less variability of the wet delay) using a random walk model with a SD of 3 mm/$\sqrt{\text{h}}$. The result indicates an insignificant difference ($< 0.05$ kg/(m$^2$·decade)) in IWV trends. Therefore in order to be able to capture any large variations of the water vapour we decided to use the large value of SD (10 mm/$\sqrt{\text{h}}$) for all 12 sites.

The ZTD estimates were updated every 5 min using the Vienna Mapping Function 1 (VMF1) (*Boehm et al.*, 2006a). They were then averaged using a time window of ± 0.5 hour, and converted to the IWV using the following procedure (*Ning et al.*, 2016a).

The ZTD is the sum of the Zenith Hydrostatic Delay (ZHD) and the Zenith Wet Delay (ZWD).

$$\ell_t^z = \ell_h^z + \ell_w^z \tag{1}$$

The ZHD for a given GPS site can be calculated:

$$\ell_h^z = \frac{2.2767 \cdot P_0}{f(\lambda, H)} \tag{2}$$





where $P_0$ is the ground pressure in hPa and

$$f(\lambda, H) = \left(1 - 2.66 \cdot 10^{-3} \cos\left(2\lambda\right) - 2.8 \cdot 10^{-7} H\right) \tag{3}$$

130 determined by $\lambda$ and $H$ which are the site latitude in degrees and the height above the geoid in m, respectively.

The ZWD is related to the IWV via the conversion factor $Q$

$$V = \frac{\ell_w^z}{Q} \tag{4}$$

where $Q$ is defined by

135 $$Q = 10^{-6} \rho_w R_w \left(k_2' + \frac{k_3}{T_m}\right) \tag{5}$$

where $\rho_w$ is the density of liquid water; $R_w$ is the specific gas constant for water vapour; $k_3$ and $k_2'$ are two constants which can be estimated from laboratory experiments; $T_m$ is a mean temperature, weighted by the wet refractivity, in units of K.

The standard data processing was performed twice using two different elevation cutoff angles

140 ($10°$ and $25°$). Thereafter we carried out several tests using different elevation-angle-dependent parameters. One northern station (SODA) and one southern station (VIS0) were selected for the test using the alternative mapping function, global mapping function (GMF), presented by *Boehm et al.* (2006b) and for the test including second-order ionospheric corrections (*Kedar et al.*, 2003) based on the International Geomagnetic Reference Field (IGRF) model (*Matteo and Morton*, 2011).

145 In addition we processed the data from 8 GPS sites again with an elevation dependent weighting function which is discussed in Sections 4.1 and 4.3. The weighting function applied was $W = \sin\left(E\right)$ where $E$ is the elevation angle. The same standard deviation ($10 \text{ mm}/\sqrt{\text{h}}$) was used for the ZTD in the random walk model.

## 2.2 Radiosonde

150 Measurements from 7 radiosonde sites (see Figure 1) were obtained from the database provided by National Oceanic and Atmospheric Administration (NOAA) (https://ruc.noaa.gov/raobs/). As seen in Table 1, the maximum distance between the GPS site and the corresponding nearby radiosonde site is around 120 km while the height difference is less than 100 m for most of the paired sites. The radiosonde data consist of vertical profiles of pressure, temperature, and humidity. We linearly

155 interpolated these profiles up to 12 km at intervals of 50 m, and integrated the absolute humidity in order to calculate the IWV. Radiosondes are at the most launched four times per day (but more common is two times per day) and the profiles are reported at the nominal time epochs 0:00, 6:00, 12:00, and 18:00 UTC. Figure 2 depicts the number of radiosonde observations obtained from each site for every year. It is clear that the launch frequency of radiosondes has changed significantly

160 over the years. For example we note that four sites (Sundsvall, Jokioinen, Jyväskylä, and Sodankylä)





have more frequent launches over the years from 2011 to 2014 while the number of launches for two sites (Landvetter and Luleå) decreases significantly over the last 10 years.

The radiosonde data have been validated by *Ning et al.* (2012) where they presented comparisons of 10-year-long time series of the atmospheric ZWD, estimated using GPS, geodetic very long baseline interferometry (VLBI), a water vapour radiometer (WVR), radiosonde observations, and the ERA-Interim for the IGS site ONSA. We note that all radiosonde sites used in this work changed the type of humidity sensor (from RS80 to RS92) late in 2005 and early in 2006. *Ning and Elgered* (2012) pointed out that since the offset due to the change of instrument will alias with the offset due to different weather conditions before and after the occurrence of change, it is very difficult to perform a reliable correction on the radiosonde-derived IWV. Meanwhile, *Ning and Elgered* (2012) found that due to similar occurrence date of changes for all radiosonde sites in the investigated region (Sweden and Finland), neglect of the offset corrections on radiosonde data resulted in insignificant impacts on the correlation coefficients between the IWV trends from the radiosonde and the GPS data.

The radiosonde data obtained from NOAA, for the site of Sodankylä, were also validated using the GRUAN corrected data where algorithms were developed to correct systematic errors in RS92 data and to derive an uncertainty estimate for each data point and each parameter (*Dirksen et al.*, 2014). Using the data covering the time period from 2011 to 2016, we found a mean IWV difference of 0.95 kg/m$^2$ with a standard deviation of 0.31 kg/m$^2$. In addition we estimated the trend in the difference between the GRUAN and the NOAA data which shows an insignificant value of 0.0021 kg/(m$^2$·decade).

Based on the discussion above and concerning the fact that potential offsets will have insignificant impacts on the resulting trends, we decided not to correct for any offsets in the radiosonde data.

### 2.3 ERA-Interim data

ERA-Interim, the reanalysis product by ECMWF, provides the IWV time series with a temporal resolution of 6 h and a horizontal resolution of about 50 km (*Berrisford et al.*, 2011). The ERA-Interim IWV were first interpolated horizontally to the GPS site using the ECMWF interpolation library (EMOSLIB, http://www.ecmwf.int). Thereafter, in order to reduce the IWV offset due to the difference between the model height and the GPS antenna height, we carried out a vertical interpolation of the ERA-Interim data to the height of the GPS antenna as follows (*Heise et al.*, 2009). If the GPS height is above the lowest ERA-Interim level, the temperature and specific humidity were linearly interpolated while pressure was logarithmically interpolated to the GPS height. If it is below the lowest level in ERA-Interim, the temperature was extrapolated using the mean temperature gradient of the three lowest layers. The pressure was calculated by stepwise application of the barometric height formula for each 20 m while the specific humidity is estimated in parallel assuming that the mean relative humidity of the two lowest ERA-Interim levels is representative for the atmosphere



below. Finally a linear temporal interpolation of the ERA-Interim data was applied to have the same temporal resolution as in the IWV time series from the GPS data (1 h).

The IWV time series obtained from ERA-Interim have been evaluated in other studies. Using ground-based GPS measurements from 99 European GNSS sites, each with a maximum time series of 14 years, *Ning et al.* (2013) found that a mean IWV difference of 0.39 kg/m$^2$ and a standard deviation of 0.35 kg/m$^2$ for the ERA-Interim−GPS comparison. The linear IWV trends estimated from the ERA-Interim data were investigated by *Bock et al.* (2014) and were compared to the ones obtained from Doppler Orbitography Radiopositioning Integrated by Satellite (DORIS) measurements at 81 global sites. The ERA-Interim data compared to the homogenized DORIS data resulted in a correlation coefficient which is larger than 0.95.

## 3   Data analysis

### 3.1   Trend estimation

Linear trends of the IWV were estimated by using a model with annual and semi-annual terms (details are described by *Nilsson and Elgered* (2008)):

$$y \quad = \quad y_0 + a_1 t + a_2 \sin(2\pi t) + a_3 \cos(2\pi t) + a_4 \sin(4\pi t) + a_5 \cos(4\pi t) \tag{6}$$

where $y$ and $t$ are the IWV and the time in years (from 1 January 1997 at UTC 0:00), respectively. The parameters $y_0$ and $a_1$ are a constant and a linear trend, respectively; $a_2$ and $a_3$ are annual component coefficients, and $a_4$ and $a_5$ are semi-annual component coefficients. All unknown coefficients are determined using the method of least squares.

In order to avoid possible differences in the estimated IWV trends due to different sampling intervals of the different techniques, a data synchronisation is necessary. This was done by using only the GPS and the ERA-Interim data acquired, from the very same hour, as the launches of the radiosondes.

### 3.2   Interventions in GPS IWV time series

Any known interventions in the GPS observations due to, e.g., antenna changes and/or radome changes, need to be corrected for before we compare the GPS-derived IWV to the ones obtained from radiosondes and ERA-Interim.

There are in total 6 GPS sites which have known hardware changes over the investigated time period. These changes are listed in Table 2. Most of the interventions are due to antenna changes while two are due to radome changes. There is one intervention which is caused by adding microwave absorbing material to the antenna at the site SPT0. The offset caused by each intervention was estimated as the mean difference in the GPS and ERA-Interim IWV difference time series before and





after the occurrence of the intervention. All estimated mean differences are also presented in Table 2

where the values vary from $-1.40$ to $+0.63$ kg/m$^2$.

In order to assess the significance of those offsets we applied the PMTred test, presented by *Ning et al.* (2016b), using the monthly mean IWV difference time series between the GPS and ERA-Interim data. Figure 3 depicts the time series of monthly mean IWV difference for three sites. The interventions for ONSA and the first one at SPT0 are easily seen by the naked eyes and both were detected

correctly by the PMTred test. However the test missed the intervention for JON0 and for other sites with interventions associated with smaller IWV offsets. This is consistent to the result shown by *Ning et al.* (2016b) where most of the interventions detected by the PMTred test have a relatively large IWV offset.

In order to carry out a correction on the GPS IWV time series for the offset caused by an inter-

vention, a reference time period needs first to be chosen. Thereafter the estimated mean differences, relative to the reference time period, were applied to the other parts of the IWV time series. We investigated the impact of using different reference time period on resulting IWV means and trends after the offset corrections for three GPS sites with two interventions in their IWV time series. We found that trends are not affected by which reference period that is chosen but the overall mean

differences compared to other techniques will. The differences in the overall means are shown in Table 3. A relatively large difference (1.4 kg/m$^2$) is seen from the site SPT0 when the last time period was used as the reference period. It is because this reference time period is short (less than 4 months). We decided to use the time period that has the smallest offset relative to the ERA-Interim IWV and has a data length longer than 1 year as the reference period for the offset corrections of the

GPS data.

Table 4 shows the mean and the standard deviation of the IWV difference between the radiosonde data and the GPS data before and after corrections for the GPS interventions. For two sites (ONSA and SPT0) the IWV mean difference change significantly after the offset correction is carried out on the GPS data. For other sites (JON0, METS, SKE0 and VAN0) the changes are insignificant. It

seems as this specific change of radome type at ONSA and the addition of the microwave absorber at SPT0 have larger impacts than receiver and antenna changes. We note that the corrections for the inconsistencies are derived from comparing GPS and ERA-Interim, but in Table 4 we compare the IWV from GPS with the radiosondes. Although radiosonde data are input to ERA-Interim, it is not granted that the correction shall have a positive impact.

## 4   Results

### 4.1   IWV comparison

The entire 20 year long IWV time series for the ONSA site are shown in Figure 4 while Figure 5 depicts comparisons of IWV estimates obtained from the different data sets. Note that one radiosonde





site can be compared to multiple GPS sites and the offset corrections were applied to the GPS sites
with interventions using the method discussed in Section 3.2. The comparisons show, as expected,
that the standard deviation of the IWV difference increases as the distance between the GPS and
the radiosonde sites becomes larger. This behaviour is not seen when ERA-Interim and GPS are
compared possibly because the ERA-Interim IWV were interpolated horizontally to the location of
the GPS site. In addition the IWV difference between the GPS 25° elevation cutoff solution and
the other two data sets gives a larger standard deviation than the corresponding ones obtained for
the GPS 10° solution. This is due to larger formal errors of the individual IWV estimates caused
by a worse satellite geometry and the reduced number of the observations when applying a higher
elevation cutoff angle.

### 4.2 The relation between the ZTD and the IWV trends

Before presenting and comparing the estimated trends in the IWV it is appropriate to assess possible trends in the parameters used in the conversion from the ZTD, estimated from the GPS data, to the IWV according to Equations (1)–(5). All estimated trends are presented in Table 5. For all the sites the observed trends in the mean temperature, $T_m$, varies from 0.29 to 0.70 K/decade which correspond to 0.11 to 0.26 %/decade if we express the trends in percentage. These relative trends shall be compared to the relative trends in the IWV which range from 0.51 to 6.28 %/decade. Therefore, the IWV trends are approximately linearly related to the ZWD trends. Actually the correlation coefficient between the ZWD and the IWV trends is 0.9991.

### 4.3 IWV trend comparison

Before comparing the IWV trends obtained from the different data sets, we calculated the corresponding trend uncertainties for the GPS data from the two different elevation cutoff angle solutions shown in Figure 6. In the top panel the trend uncertainties were obtained using the formal error of the individual IWV estimates assuming a white noise behaviour. As a result the trend uncertainties obtained from the two solutions are very small ($\sim$0.015 kg/(m$^2$·decade) for the  25° solution and $\sim$0.005 kg/(m$^2$·decade) for the 10° solution) using a time period of 20 years. This type of uncertainty, however, only indicates how the estimated trend differs from what would be expected if there is no other errors, or deviations, in the IWV data compared to the model. Actually the estimated IWV trends have rather large uncertainties caused by the true short term variation (the natural variability of the weather) which is not described by the model, i.e., the deviations from the model are correlated in time. In order to calculate the trend uncertainty after taking these variations into account, we used a model presented by *Nilsson and Elgered* (2008). These uncertainties are shown in the bottom panel of Figure 6 where the two solutions with different elevation cutoff angles show similar values varying between 0.20 and 0.25 kg/(m$^2$·decade).



### 4.3.1 The impact of different elevation cutoff angles

The GPS-derived IWV trends for the two solutions using different elevation cutoff angles (and the

standard data processing) and the synchronised trends from the ERA-Interim and the radiosonde data are presented in Table 6 where the estimated trends and the corresponding uncertainties (after taking the short term variation of the water vapour into account), are given before and after the plus-minus sign ($\pm$), respectively. Offset corrections were implemented for all GPS sites with interventions. An overall result is that all estimated trends are positive (except one with a very small negative value).

The trends from the ERA-Interim show a smaller variation (from 0.07 to 0.53 kg/(m$^2$·decade)) compared to those from GPS and radiosonde data. It is clear that the estimated IWV trends are comparable to the trend uncertainties, varying from 0.20 to 0.26 kg/(m$^2$·decade), for all techniques. The similar values of the trend uncertainties are expected due to the fact that all data sets were acquired during the same time period and weather conditions. In addition we observe that the trend differences

between the GPS data and the radiosonde data show no clear correlation to the site separation.

Table 6 also presents the mean trends (from 0.34 to 0.39 kg/(m$^2$·decade)) and SD of the trends (from 0.11 to 0.21 kg/(m$^2$·decade)) over all sites. The values given by *Ning and Elgered* (2012) were 0.01, 0.05, and 0.03 kg/(m$^2$·decade) for the mean trends obtained for the GPS elevation 10° solution, the elevation 25° solution, and radiosondes with SD of 0.33, 0.41, and 0.44 kg/(m$^2$·decade),

respectively. The less consistence of the trends (large SD values) shown in *Ning and Elgered* (2012) is because of the shorter time period of the data (14 years) meaning that the trend shows more sensitivity to deviations from the model especially in the beginning and in the end of the selected time series (*Nilsson and Elgered*, 2008).

Figure 7(a) depicts the comparison of the trends between the radiosonde data and the ones given

by the GPS data for the two different elevation cutoff angle solutions. Similar correlation coefficients (0.71 and 0.74) are observed for the 10° and 25° solutions, respectively. In addition, the 25° solution gives a slightly lower root-mean-square (RMS) difference (0.15 kg/(m$^2$·decade)). The GPS trends were also compared to the ones obtained from the ERA-Interim data (see Figure 7(b)). A higher correlation coefficient (0.9) and a lower RMS difference (0.09 kg/(m$^2$·decade)) are seen for the

elevation 25° solution than the ones (0.53 and 0.18 kg/(m$^2$·decade)) for the 10° solution.

As discussed in Section 3.2 we used ERA-Interim as the reference data for the offset corrections of interventions in the GPS data. It should be noted that inconsistencies may also exist in the ERA-Interim IWV (*Dee and Uppala*, 2008), it is recommended both by *Vey et al.* (2009) and *Ning et al.* (2016b) that the offset corrections using the ERA-Interim data as the reference need to be further

validated and confirmed using other reference data, e.g., the data from nearby GPS site and/or a nearby geodetic VLBI site. In this work only the IGS site ONSA has a nearby VLBI telescope and the offset correction we applied for ONSA was confirmed by other studies, i.e., *Ning et al.* (2013) and *Ning et al.* (2016b). In order to investigate the possible impact of the unvalidated intervention corrections, we compared the IWV trends again but now using only the 8 GPS sites without interven-



tions and ONSA. The results are shown in Figure 8. The correlations between the trends are almost
the same as the previous result based on all sites.

### 4.3.2   The impact of additional elevation-angle-dependent parameters

Insignificant differences in the estimated IWV trends are observed when using different mapping
functions and the implementation of the second-order ionospheric corrections in the GPS data pro-
cessing. The choice of mapping function has a very small impact for both sites (SODA and VIS0).
The differences for elevation 10° and 25° solutions are less than 0.03 kg/(m$^2$·decade) and 0.005
kg/(m$^2$·decade), respectively. Even smaller differences ($< 0.007$ kg/(m$^2$·decade)) are seen when us-
ing second-order ionospheric corrections in the data processing.

We compared the GPS-derived IWV trends with a data weighting to the trends obtained from
radiosondes and ERA-Interim. The results are shown in Figure 9 where the trend correlations and
the RMS differences are in general slightly worse than using the GPS data without the elevation
dependent weighting (see Figures 7 and 8).

### 5   Conclusions

We have processed 20 years of GPS data acquired from 13 GNSS sites in Sweden and Finland using
the two different elevation cutoff angles of 10° and 25°. We also carried out several tests assessing
the impact of three additional elevation-angle-dependent parameters: different mapping functions,
inclusion of the second-order ionospheric corrections, and applying elevation-dependent weighting
of the observations. The GPS-derived IWV were compared to the ones obtained from the radiosonde
data at 7 nearby ($< 120$ km) sites and the IWV given by the ERA-Interim data.

We show that due to the larger formal errors of the individual IWV estimates a larger standard
deviation is seen for the individual estimates of the IWV difference between the GPS elevation
25° solution and the other two techniques. On the other hand the larger formal error of the individual
IWV estimates is not the limiting factor for the uncertainty of the estimated IWV trends. We obtain
similar correlation coefficients and RMS differences when comparing the trends obtained from the
GPS elevation cutoff angle solutions at 25° and 10° with the trends obtained from the radiosonde
data. A higher correlation and lower RMS difference are obtained for the GPS 25° solution com-
pared to the 10° solution when the two are compared to the IWV trends from the ERA-Interim data.
The results demonstrate that the selection of mapping function and the use of the second-order iono-
spheric corrections are not critical when using GPS-derived IWV for applications when estimating
linear trends over decades. Moreover elevation dependent weighting does not improve the agree-
ments and gives the same relative performance when comparing the solutions with the two different
cutoff angles.



The results show that using different elevation cutoff angles is a valuable diagnostic tool that can
be used for validation purposes and detection of possible site problems, such as multipath impacts.

When we use the GPS data to monitor the long-term change in the IWV, e.g., as linear trends, it is
recommended to apply at least two different elevation cutoff angles in the data processing. Ideally the
IWV trends obtained from the two significantly different cutoff angle elevation solutions should be
the same if there is no significant long-term changes in the multipath impacts, or any other elevation
dependent phenomena that affects the observations.

Compared to our previous study (*Ning and Elgered*, 2012) we find that the estimated trends for
the different sites now are more consistent. For the 25° elevation cutoff angle the mean and standard
deviation are for the 20 years 0.35 and 0.18 kg/m$^2$ compared to 0.08 and 0.41 kg/m$^2$ for the 14
years of data. Both in this study and the previous one we find that for no site the estimated trend
becomes significantly worse when the 25° cutoff angle was used. In fact when compared to the trends

obtained from the radiosonde and the ERA-Interim (only in this study) data a higher correlation
coefficient and a lower RMS difference are seen for the 25° solution. Therefore the high cutoff angle
is desired to be used in order to estimate the long-term trend in the IWV. Meanwhile, as suggested
in (*Ning and Elgered*, 2012) it is important to carry out similar studies for other sites and especially
from areas with different climates since the optimum cutoff angle (25°) for our investigated area

may be different for GPS sites in different electromagnetic environments and sites at lower latitudes,
where the distribution of observations as a function of elevation angle is different.

*Acknowledgement.* We would like to thank Stefan Heise for providing the ERA-Interim data. We also thank
the National Land Survey of Finland for making the FinnRef GPS data available. The map in Figure 1 was
produced using the Generic Mapping Tools (*Wessel and Smith*, 1998).



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



**Table 1.** GPS sites and the nearby radiosonde sites, sorted by decreasing latitude.

| | GPS site | Long. | Lat. | Height[a] | RS site | Long. | Lat. | Height[a] | Distance | Height diff. |
|---|---|---|---|---|---|---|---|---|---|---|
| Acronym | Name | [°E] | [°N] | [m] | Name | [°E] | [°N] | [m] | [km] | [m] |
| SODA | Sodankylä | 26.39 | 67.42 | 279 | Sodankylä | 26.65 | 67.37 | 158 | 12 | 121 |
| OVE0 | Överkalix | 22.77 | 66.31 | 200 | Luleå | 22.13 | 65.55 | −6 | 90 | 206 |
| SKE0 | Skellefteå | 21.05 | 64.88 | 59 | Luleå | 22.13 | 65.55 | −6 | 90 | 65 |
| KIVE | Kivetty | 25.70 | 62.82 | 198 | Jyväskylä | 25.67 | 62.40 | 114 | 47 | 84 |
| SUN0 | Sundsvall | 17.66 | 62.23 | 7 | Sundsvall | 17.47 | 62.53 | −20 | 35 | 27 |
| OLKI | Olkiluoto | 21.47 | 61.24 | 12 | Jokioinen | 23.50 | 60.82 | 84 | 119 | −72 |
| TUOR | Tuorla | 22.44 | 60.42 | 41 | Jokioinen | 23.50 | 60.82 | 84 | 73 | −43 |
| METS | Metsähovi | 24.40 | 60.22 | 76 | Jokioinen | 23.50 | 60.82 | 84 | 83 | −8 |
| VAN0 | Vänersborg | 12.07 | 58.69 | 135 | Landvetter | 12.30 | 57.67 | 119 | 114 | 16 |
| JON0 | Jönköping | 14.06 | 57.75 | 227 | Landvetter | 12.30 | 57.67 | 119 | 105 | 108 |
| SPT0 | Borås | 12.89 | 57.72 | 185 | Landvetter | 12.30 | 57.67 | 119 | 37 | 66 |
| VIS0 | Visby | 18.37 | 57.65 | 55 | Visby | 18.35 | 57.65 | 20 | 1 | 35 |
| ONSA | Onsala | 11.93 | 57.40 | 9 | Landvetter | 12.30 | 57.67 | 119 | 37 | −110 |

[a]The heights are referenced to the mean sea level.

**Table 2.** Known GPS station-related changes and the corresponding estimated mean IWV differences caused by the intervention.

| Site | Date | Type of change | Mean difference for elevation 10° [kg/m$^2$] | PMTred test | Mean difference for elevation 25° [kg/m$^2$] | PMTred test |
|---|---|---|---|---|---|---|
| JON0 | 2002-08-23 | Antenna | −0.16 | NONE | −0.32 | NONE |
| METS | 2010-08-19 | Antenna | 0.29 | NONE | −0.01 | NONE |
| METS | 2013-06-28 | Antenna | −0.19 | NONE | −0.22 | NONE |
| ONSA | 1999-02-02 | Radome | 0.63 | 1999-02 | −1.60 | 1999-02 |
| SKE0 | 2003-09-27 | Antenna | −0.11 | NONE | −0.04 | NONE |
| SKE0 | 2008-03-14 | Antennae | −0.27 | NONE | −0.24 | NONE |
| SPT0 | 2007-06-09 | Absorber | −0.50 | 2007-05 | 0.01 | NONE |
| SPT0 | 2016-08-23 | Antenna | −0.35 | NONE | −1.40 | 2015-06 |
| VAN0 | 2003-03-30 | Radome | −0.16 | NONE | 0.29 | NONE |



**Table 3.** Differences in IWV mean values due to different selections of the reference time period for intervention corrections

| GPS site | Reference 1[a] | Reference 2[b] | Reference 3[c] | GPS [c] | ERA-Interim | Radiosonde |
|---|---|---|---|---|---|---|
| | \[kg/m$^2$\] | \[kg/m$^2$\] | \[kg/m$^2$\] | \[kg/m$^2$\] | \[kg/m$^2$\] | \[kg/m$^2$\] |
| **Solution for an elevation cutoff angle at 10°** | | | | | | |
| METS | 12.90 | 13.19 | 12.99 | 12.96 | 12.87 | 13.32 |
| SKE0 | 11.70 | 11.59 | 11.32 | 11.59 | 11.40 | 12.14 |
| SPT0 | 13.58 | 13.08 | 12.73 | 13.44 | 13.45 | 13.47 |
| **Solution for an elevation cutoff angle at 25°** | | | | | | |
| METS | 13.11 | 13.12 | 12.91 | 13.08 | 12.87 | 13.32 |
| SKE0 | 11.12 | 11.08 | 10.84 | 11.05 | 11.40 | 12.14 |
| SPT0 | 12.02 | 12.04 | 10.62 | 12.00 | 13.45 | 13.47 |

[a] Use the first part of the time period without interventions as the reference.

[b] Use the middle part of the time period without interventions as the reference.

[c] Use the last part of the time period without interventions as the reference.

[d] The GPS mean value before the corrections for the interventions.



**Table 4.** The IWV comparison between radiosonde data and the GPS data before and after the corrections for the interventions in the GPS time series.

| | Before corrections | | After corrections | |
|---|---|---|---|---|
| | Elevation 10° solution − Radiosonde | | | |
| GPS site | Mean difference | Standard deviation | Mean difference | Standard deviation |
| | [kg/m$^2$] | [kg/m$^2$] | [kg/m$^2$] | [kg/m$^2$] |
| JON0 | −0.63 | 2.12 | −0.55 | 2.11 |
| METS | 0.09 | 1.86 | 0.32 | 1.86 |
| ONSA | 0.40 | 1.62 | 0.55 | 1.60 |
| SKE0 | 0.19 | 1.93 | 0.30 | 1.92 |
| SPT0 | −0.37 | 1.23 | 0.12 | 1.19 |
| VAN0 | −0.22 | 2.37 | −0.14 | 2.36 |
| | Elevation 25° solution − Radiosonde | | | |
| JON0 | −0.48 | 2.20 | −0.32 | 2.20 |
| METS | 0.22 | 1.99 | 0.26 | 1.98 |
| ONSA | 0.16 | 1.81 | −0.21 | 1.74 |
| SKE0 | −0.36 | 2.07 | −0.28 | 2.06 |
| SPT0 | −1.43 | 1.38 | −1.44 | 1.36 |
| VAN0 | −0.82 | 2.47 | −0.96 | 2.46 |





**Table 5.** The IWV trends from the GPS elevation $10°$ and the trends in parameters used in the conversion from the ZTD to the IWV. The sites are sorted by decreasing latitude.

| GPS site | ZTD | ZHD | Pressure $P_0$ | ZWD | Mean temperature $T_m$ | IWV |
|---|---|---|---|---|---|---|
| | [mm/decade] | [mm/decade] | [hPa/decade] | [mm/decade] | [K/decade] | [kg/(m$^2$·decade)] |
| SODA | 2.99 | 0.06 | 0.03 | 2.93 | 0.53 | 0.45 |
| OVE0 | 2.82 | 1.65 | 0.73 | 1.16 | 0.70 | 0.20 |
| SKE0 | 3.66 | 1.83 | 0.81 | 1.82 | 0.50 | 0.29 |
| KIVE | 2.47 | 0.77 | 0.34 | 1.70 | 0.52 | 0.27 |
| SUN0 | 2.48 | 1.42 | 0.63 | 1.05 | 0.43 | 0.18 |
| OLKI | 7.46 | 1.98 | 0.87 | 5.47 | 0.47 | 0.87 |
| TUOR | 5.21 | 2.00 | 0.88 | 3.21 | 0.59 | 0.53 |
| METS | 4.72 | 2.19 | 0.96 | 2.53 | 0.52 | 0.41 |
| VAN0 | 2.88 | 1.16 | 0.51 | 1.71 | 0.35 | 0.28 |
| JON0 | 2.86 | 1.47 | 0.65 | 1.39 | 0.31 | 0.23 |
| VIS0 | 1.89 | 1.48 | 0.65 | 0.42 | 0.32 | 0.07 |
| SPT0 | 2.38 | 1.39 | 0.61 | 0.99 | 0.32 | 0.17 |
| ONSA | 2.39 | 1.12 | 0.49 | 1.27 | 0.29 | 0.21 |



**Table 6.** The estimated IWV trends from all data sets.

| GPS site | Radiosonde site | Distance [km] | Number of paired observations | Trend [kg/(m²·decade)] | | | |
|----------|-----------------|---------------|-------------------------------|-------------------------|-----------|------------|-------------|
| | | | | GPS 10° | GPS 25° | Radiosonde | ERA-Interim |
| VIS0 | Visby | 1 | 11007 | 0.07±0.24 | −0.05±0.24 | 0.08±0.25 | 0.07±0.23 |
| SODA | Sodankylä | 12 | 10756 | 0.45±0.21 | 0.50±0.23 | 0.29±0.22 | 0.34±0.21 |
| SUN0 | Sundsvall | 35 | 15338 | 0.18±0.23 | 0.22±0.23 | 0.40±0.24 | 0.30±0.23 |
| SPT0 | Landvetter | 37 | 11436 | 0.17±0.23 | 0.25±0.23 | 0.32±0.24 | 0.30±0.23 |
| ONSA | Landvetter | 37 | 11420 | 0.21±0.25 | 0.23±0.26 | 0.32±0.24 | 0.34±0.24 |
| KIVE | Jyväskylä | 47 | 9947 | 0.27±0.22 | 0.28±0.23 | 0.10±0.23 | 0.26±0.22 |
| TUOR | Jokioinen | 73 | 11644 | 0.53±0.26 | 0.66±0.26 | 0.79±0.25 | 0.53±0.25 |
| METS | Jokioinen | 83 | 12366 | 0.41±0.24 | 0.40±0.24 | 0.65±0.24 | 0.44±0.24 |
| OVE0 | Luleå | 90 | 10805 | 0.20±0.20 | 0.50±0.20 | 0.41±0.23 | 0.44±0.20 |
| SKE0 | Luleå | 90 | 10926 | 0.29±0.23 | 0.37±0.23 | 0.42±0.23 | 0.40±0.23 |
| JON0 | Landvetter | 105 | 11636 | 0.23±0.23 | 0.24±0.23 | 0.31±0.24 | 0.28±0.23 |
| VAN0 | Landvetter | 114 | 11584 | 0.28±0.23 | 0.40±0.24 | 0.33±0.24 | 0.33±0.23 |
| OLKI | Jokioinen | 119 | 10655 | 0.87±0.26 | 0.50±0.26 | 0.68±0.25 | 0.38±0.25 |
| Mean trend | | | | 0.32 | 0.35 | 0.39 | 0.34 |
| Standard deviation | | | | 0.21 | 0.18 | 0.21 | 0.11 |



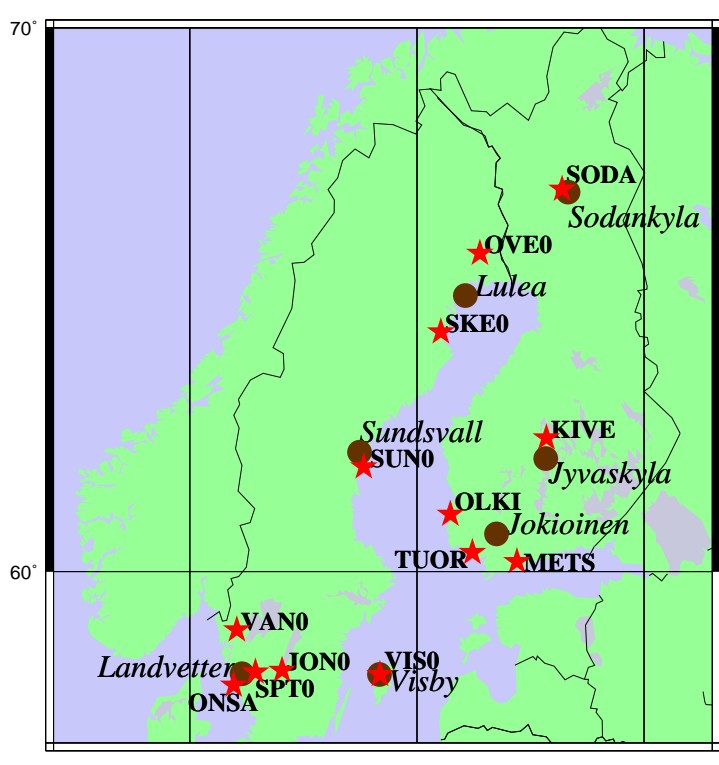

**Figure 1.** The 13 GPS sites (red stars) and the 7 radiosonde sites (brown dots).



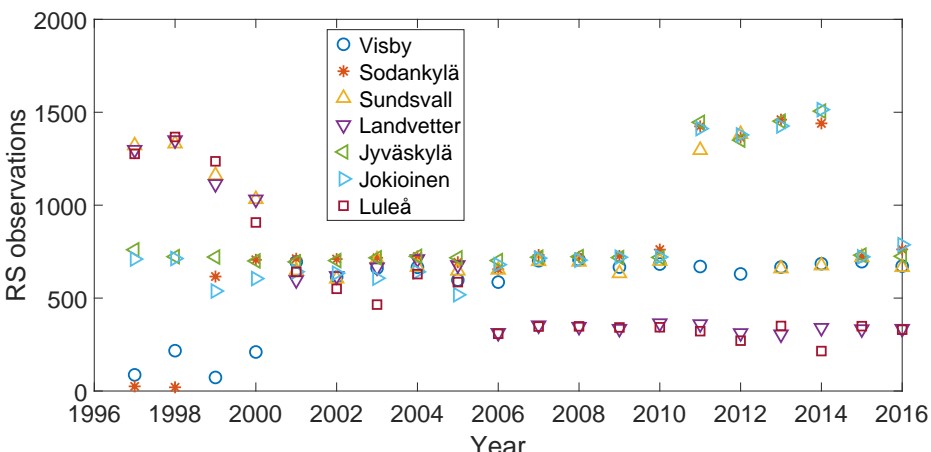

**Figure 2.** The number of radiosonde launches per year.





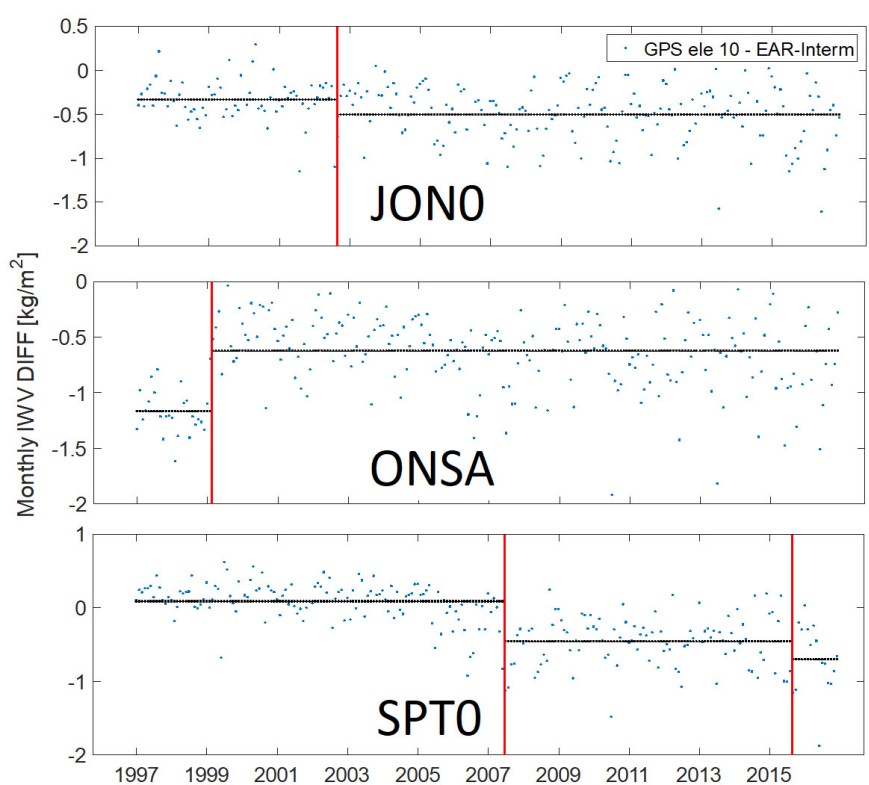

**Figure 3.** Time series of the monthly mean IWV difference (GPS– ERA-Interim) for three sites: JON0, ONSA, and SPT0. Dark lines are the mean of IWV difference, and red lines indicate the date of the interventions.




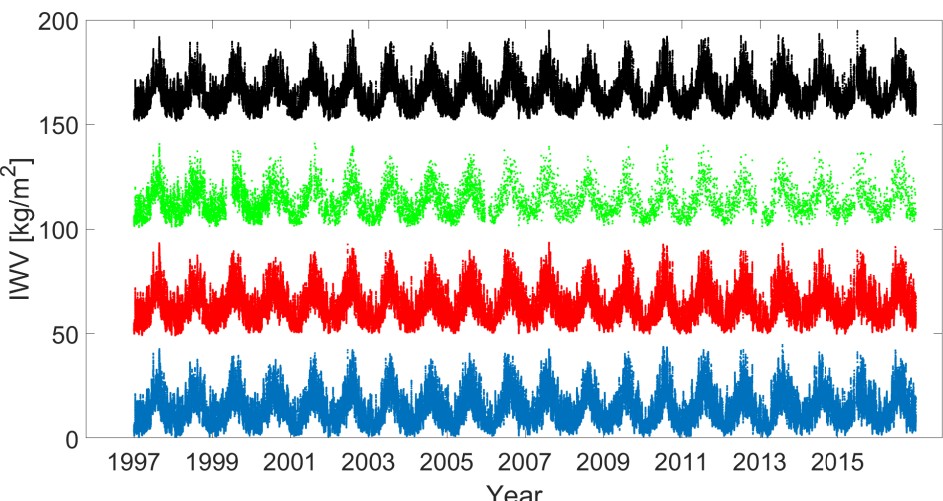

**Figure 4.** Time series of the IWV derived from the different techniques at the IGS site ONSA. Note that offsets were added to the time series from the GPS elevation 25° cutoff angle solutions (red)+50 kg/m², the radiosonde observations at Landvetter 37 km away (green)+100 kg/m², and ERA-Interim data (black)+150 kg/m². No offset was added to the time series of the GPS elevation 10° cutoff angle solution (blue).

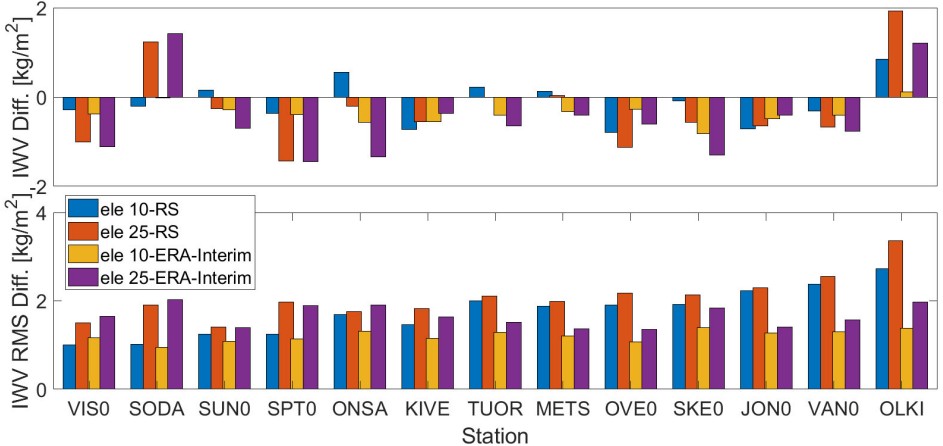

**Figure 5.** The mean (top) and the root-mean-square (bottom) of the IWV differences for the two different elevation cutoff angles. The GPS sites (from left to right) are sorted by increasing distance to the radiosonde site.




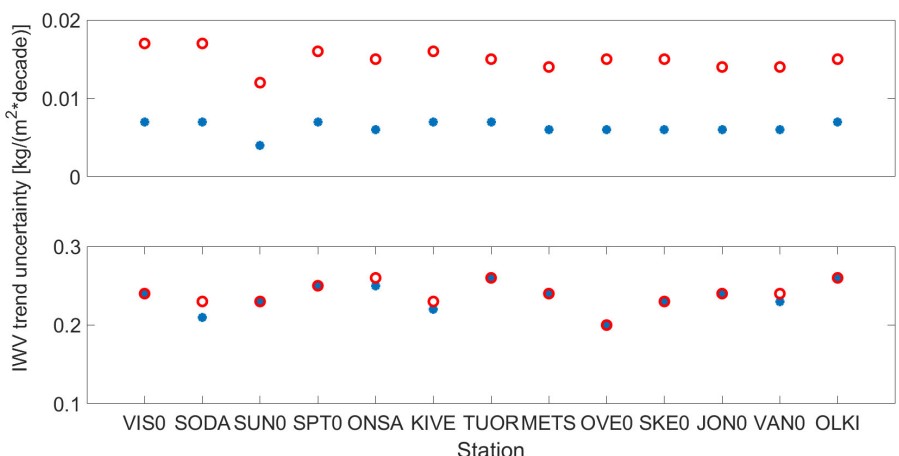

**Figure 6.** The uncertainties of the IWV trends obtained using the formal error of the GPS estimates and assuming a white noise behaviour (top), and after rescaling and taking the temporal correlation of the IWV into account (bottom). Open red circles and filled blue circles denote an elevation cutoff angle of 25° and 10°, respectively.

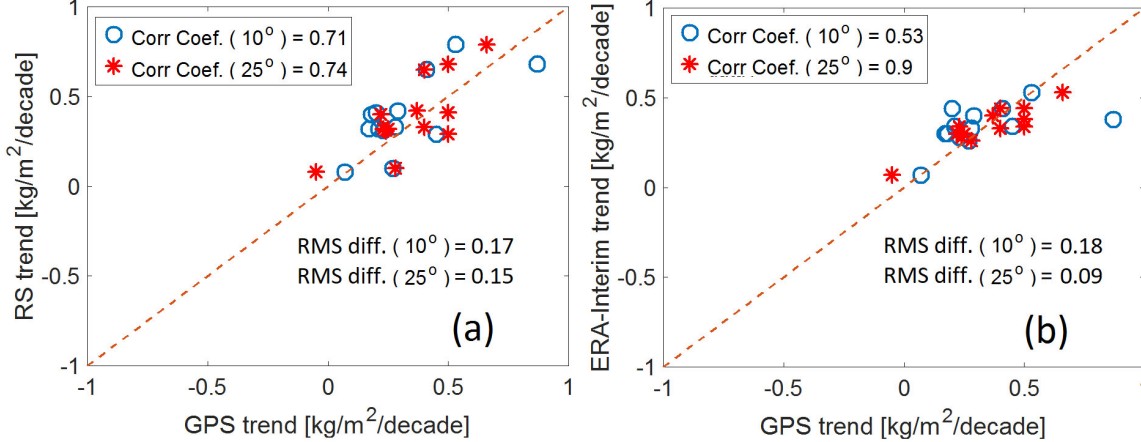

**Figure 7.** Correlations between the IWV trends from the GPS and the radiosonde data (a), and the ERA-Interim data (b) for 10° and 25° elevation cutoff angles. The dashed lines show the perfect agreement.



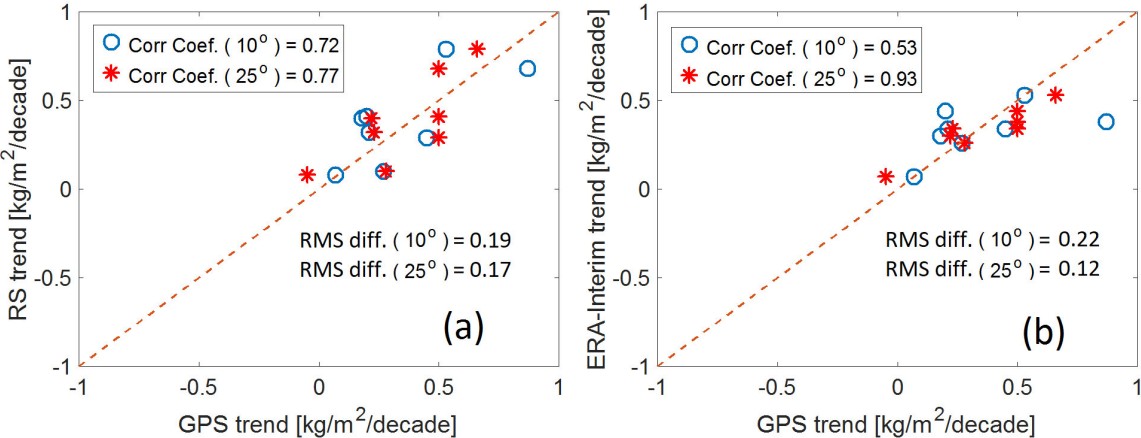

**Figure 8.** Same as Figure 7 but only including the 8 GPS sites with no interventions and the site ONSA.

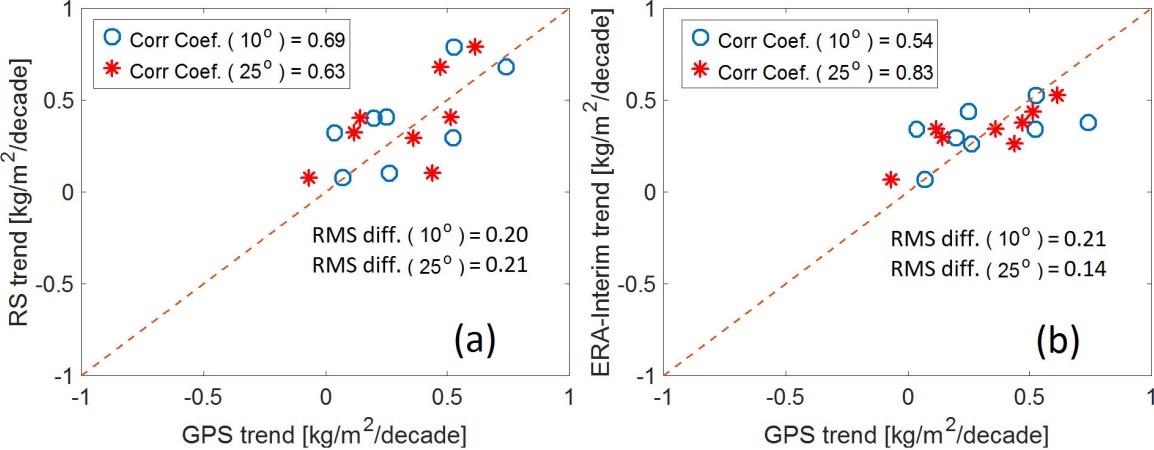

**Figure 9.** Same as Figure 8 but the GPS-derived trends were obtained from the data processing applying an elevation-angle-dependent weighting.