# Peer review of "Trends in the atmospheric water vapour estimated from GPS data for different elevation cutoff angles"

_Atmospheric Measurement Techniques, 2018_

## Referee Comment (RC1) · Anonymous Referee #1 · 20 Dec 2018

The paper describes for a small sample of high-latitude (> 55°) GPS sites trends in atmospheric water vapour calculated when using two different elevation cut-off angles, in comparison with "co-located" radiosonde time series and IWV time series calculated from ERA-Interim at those sites. A more or less identical study has been described by the same authors in Ning & Elgered (2012), based on the same stations, but without the inclusion of the ERA-Interim IWV time series.

MAJOR COMMENTS

This observation immediately brings me to the largest drawback of the current study: the lack of scientific novelty. Although the manuscript is well written and the analysis

is really done very carefully and detailed, taking into account all the related issues, the added value of this study with respect to the earlier Ning & Elgered (2012) results is really minor. Moreover, the authors also do not argue enough why there is a need for this follow-up study and what the major improvements/changes in the methodology of this study are, compared to the earlier one. This is rather disappointing. In the conclusions of both papers, the authors themselves give a hint of what a meaningful follow-up study would be: "It is important to carry out similar studies for other sites and especially from areas with different climates. Furthermore, the optimum cutoff angle (25 ) for the trend estimation may be different for GPS sites in different electromagnetic environments and sites at lower latitudes, where the distribution of observations as a function of elevation angle is different" (Ning et al., 2012). Especially taking into account that one of the authors has already done IWV time series analyses on a larger dataset (Ning et al., 2013, Ning et al., 2016b), it should be argued strongly in the manuscript why their suggestion has not been followed in this study.

The rather small dataset of only 13 sites , with rather similar IWV field properties I assume (at least belonging to a similar geographical region), weakens the significance of the analysis and possible conclusions: if correlations are based only on 13 points (Figure 7) or even on 8 points (Figures 8 & 9), is it, from a statistical point of view, meaningful to ascribe differences in the correlation coefficients between trend estimates to different elevation cut-off angles used, differences in datasets (RS vs ERA-Interim), elevation-angle-dependent weighting? Are the differences in the correlation coefficients statistically significant to draw the conclusion that "In fact when compared to the trends obtained from the radiosonde and the ERA-Interim (only in this study) data a higher correlation coefficient and a lower RMS differences are seen for the $25°$ solution. Therefore the high cutoff angle is desired to be used in order to estimate the long-term trend in the IWV" (page 12, lines 379-380)?

A last major concern is that the manuscript needs more interpretation for some fundamental statements or issues. I will give some more examples in my minor comments

here below, but I already want to highlight two of them here: first, in Section 4.1 ("IWV intercomparison"), the discussion immediately starts with describing the standard deviations of the differences between IWV GPS and IWV RS/ERA-Interim, but no description or explanation is given of the IWV biases between GPS and RS, GPS and ERA-Interim, GPS elev 10 and GPS elev 25 (and RS and ERA-Interim, see below). Secondly, in Section 4.3 ("IWV trend comparison") , describing Fig. 6: give an explanation about the difference of the trend estimations between the 10° and 25° solutions for (i) both white noise + time correlated behavior separately and (ii) between the white noise + time correlated behavior (if there are any, difficult to see on lower panel of Fig. 6 for majority of the sites).

MINOR COMMENTS

* Given the fact that the sample of sites is rather geographically constrained, I would add "at Swedish and Finnish sites" (or "at high-latitude sites", or "in a sample of Scandinavian sites") to the title.

* Page 3, lines 65-67: here, you should really point out what the added value of this study is compared to Ning and Elgered (2012) and why there is a need for this follow-up study.

* Page 5, lines 141-144: why do you only select 2 stations for the test using the alternative mapping functions and for the test including second-order ionospheric corrections? Will this be significant enough to draw conclusions? What will be the sample bias for the conclusions?

* Page 5, lines 145-148: again, why did you not consider the entire dataset of 13 sites for processing the data with an elevation dependent weighting function?

* Page 6-7: Section 2.3: here, you should add that radiosonde data are assimilated in ERA-Interim and that these are not completely independent datasets, when used as references (as is done in this study). This was a component that I was missing

in the analysis: how do the ERA-Interim and radiosonde IWV measurements/trends compare? Is there a need for having two reference datasets in this study?

\* Page 6, line 186: which version of ERA-Interim are you using? I thought the horizontal resolution of ERA-Interim is about 80 km? Please specify!

\* Page 6, line 188: please give some details about the horizontal interpolation applied by EMOSLIB: weighted? How many pixels are considered?

\* Page 6, lines 188-189: please give some numbers of the difference between the model height and GPS antenna height for your sample of sites. This might enlighten why you use an altitude correction between GPS and ERA-Interim, but not between GPS and RS.

\* Page 7, lines 199-206: if you want to mention other studies using IWV time series obtained from ERA-Interim, please use also the most recent ones and certainly those in this Special Issue: Parracho et al., ACP 2018, Pacione et al., AMT 2017, and to a lesser extent Berckmans et al., ACP 2018, Van Malderen et al., ACP 2018 (not peer-reviewed yet).

\* Page 7, lines 218: speaking about the launch times of radiosondes: did you found different results when treating the 0h (nighttime) and 12h (daytime) measurements separately in your analysis, also when compared with GPS and ERA-Interim?

\* Page 8, lines 231-232: did you try the other way around: have you applied the PMTred test on the differences time series between GPS and ERA-Interim and did you find offsets which could not be linked to interventions?

\* Page 8, lines 243-244: "We found that trends are not affected by which reference period that is chosen" → please quantify, give a number here.

\* Page 9, line 282: what is the correlation between the ZTD and IWV trends?

\* Page 10, lines 311-314: higher trend values are obtained here, i.e. for the longer time

period, than in the Ning and Elgered (2012) study. This might be a little out of scope of this paper, but does this fact means that there is an enhanced moistening in your sample of sites? And if so, what might be the cause for this?

* Page 10, lines 323-325: "A higher correlation coefficient (0.9) and a lower RMS difference (0.09 kg/(m$^2$ decade)) are seen for the elevation 25$^\circ$ solution than the ones (0.53 and 0.18 kg/(m$^2$ decade)) for the 10$^\circ$ solution". What is causing those differences, to your opinion? And are these differences significant enough to conclude that the 25$^\circ$ solution should be used for IWV trend estimation?

* Page 11, lines 345-347: what is the reason for the slightly worse trend correlations and RMS differences when using GPS data with elevation dependent weighting? And are the differences significant?
* * *

---

## Referee Comment (RC2) · Anonymous Referee #2 · 30 Jan 2019

Referee report of manuscript amt-2018-279 "Trends in the atmospheric water vapour estimated from GPS data for different elevation cutoff angles" by Tong Ning and Gunnar Elgered

General comments

This manuscript reports on the impact of GPS data elevation cutoff angles and other processing options on IWV trend estimates in Scandinavia. This work is very similar in the concepts, ideas, and methods to a previous publication by the authors (Ning and Elgered, IEEE, 2012). The processing options are evaluated based on the correlation coefficients (and RMS differences) between the GPS trends and radiosonde and ERA-

Interim trends at 13 GPS sites. Compared to their earlier paper, differences are with the length of the GPS series (20 yrs compared to 14), the use of ERA-Interim as a second validation dataset, and the test of other processing options (mapping functions, correction of higher order ionospheric effects, and elevation-depending weighting). The longer time series and the use of a second validation dataset yield more statistical confidence into the new results. However, the conclusions remain unchanged and the authors still recommend using a 25° cutoff angle rather than 10° (though only these two cutoff angles are tested in this study) and note that the other processing options that were tested are insignificant. Little new knowledge is brought actually compared to the earlier paper.

One or both of following directions should be considered to increase the relevance of this study: 1) investigate the reasons of the different trend values found for the different cutoff angles by inspecting carefully the differences in the estimated IWV time series (are the differences due to drifts in the time series? If yes what could be the reasons? Are they due to multiple offsets due e.g. to documented or undocumented equipment changes?); 2) extend the study to other regions/climates where the sensitivity to cutoff angle and/or the other processing options tested here might be different. One can note that this was suggested by the authors themselves in this manuscript and in their previous publications.

Detailed comments

Given the small number of GPS sites used in this study, the statistical significance of the computed correlations and RMS differences is rather small (though not quantified). It seems thus hazardous to draw general conclusions on the choice of the cutoff angle. More insightful analysis is indeed required to convince the readers to use a 25° cutoff angle for trend estimates, especially since the general tendency in the GPS community is to use lower cutoff angles (typically between 3° and 10°) and the IWV comparison (GPS vs. radiosondes and ERA-Interim) shows that the biases and standard deviations increase when the cutoff angle is increased. Moreover, it should be recognised that

trend estimates are sensitive to small changes in the mean bias and extremes at the beginning and end of the time series, and thus conclusions based on trend estimates can be tricky.

A case by case analysis may help understanding the reasons why trend estimates change between 10 and 25° cutoff angles at some of the 13 sites investigated (e.g. OVE0, OLKI) and may strengthen the conclusions.

Why are the data not reprocessed for all the cutoff angles, e.g. between 5° and 30° or more, as in Ning and Elgered, 2012?

Why is only antenna, radome, and microwave absorber changes considered as GPS interventions? Did the authors check that receiver changes do generate breakpoints?

Moreover, in several places in the manuscript, the breakpoints in the GPS series not explained by antenna, radome, and microwave absorber changes are attributed to environment changes resulting in changes in multipath. Firstly, this attribution may be wrong because receiver changes are ignored. Secondly, the attribution to multipath is pure speculation as no additional observation/data/information is provided to support this hypothesis. After the receiver changes are checked, I recommend to call the remaining breakpoints 'unknown' or 'undocumented' unless a true multipath diagnostic is provided.

Regarding the choice of period for correction of the GPS interventions, the one with the smallest bias compared to ERA-Interim might not be the best choice since ERA-Interim itself may contain biases. Why didn't the authors use the more recent period following their previous work (Ning and Elgered, 2012)?

The mapping function test and second-order ionospheric corrections performed on only two sites are not significant and don't add anything to the study as the impact of these parameters is known from past studies to be small in the study area. If to be mentioned, they may simply be included in the discussion section along with the elevation weighting

results (Fig. 9 unnecessary).

Use statistical tests to assess the significance of trend estimates and differences.

The authors recommend to compare the trends computed from two different cutoff angle elevation solutions. What should be done when they yield significantly different values? Data from ERA-Interim and radiosondes should be intercompared and checked for inhomogeneities as well. Why didn't the authors use a homogenized radiosonde dataset? (e.g. Dai et al., 2011)

Dai, A., J. Wang, P. W. Thorne, D. E. Parker, L. Haimberger, and X. L. Wang (2011), A new approach to homogenize daily radiosonde humidity data, J. Clim., 24, 965–991.

Table 5: are the ZHD trends significant? They are not discussed in the text. Figure 4: is this figure really useful?

Figure 8, 9: unnecessary figures, but the statistics could be included in a Table.

---

## Author Comment (AC1) · 15 Apr 2019

The comment was uploaded in the form of a supplement:
https://www.atmos-meas-tech-discuss.net/amt-2018-279/amt-2018-279-AC1-supplement.pdf

---

## Author Comment (AC2) · 15 Apr 2019

**For the best experience, open this PDF portfolio in Acrobat X or Adobe Reader X, or later.**

**Get Adobe Reader Now!**